The mechanism analysis of exogenous melatonin in limiting pear fruit aroma decrease under low temperature storage

Wei Shuwei 1
Jiao Huijun 1
Wang Hongwei 1
Ran Kun 1
Dong Ran 1
Dong Xiaochang 1
Yan Wenjing 2
Wang Shaomin 1 sdipwsm@163.com
1 Shandong Institute of Pomology , TaiAn , China
2 Shandong Agricultural University , TaiAn , China
Tan Dunxian
Electronic publication date: 2022 Oct 14
Publication date: 2022
Volume: 10
Electronic Location ID: e14166
Received 2022 Apr 5; Accepted 2022 Sep 12
Copyright: © 2022 Wei et al.
Copyright year: 2022
Copyright holder: Wei et al.
License: This is an open access article distributed under the terms of the Creative Commons Attribution License, which permits unrestricted use, distribution, reproduction and adaptation in any medium and for any purpose provided that it is properly attributed. For attribution, the original author(s), title, publication source (PeerJ) and either DOI or URL of the article must be cited.
License URL: https://creativecommons.org/licenses/by/4.0/

Keywords: Pear, Melatonin, Aroma

Funding: China Agriculture Research System of MOF and MARA CARS-28-36 Shandong Province Agricultural Improved Seed Project 2019LZGC008 National Natural Science Foundation of China 31801845 Natural Science Foundation of Shandong Province ZR2019BC075, ZR2020MC141, ZR2021MC177 The research was supported by the grant from the China Agriculture Research System of MOF and MARA (CARS-28-36), the Shandong Province Agricultural Improved Seed Project (2019LZGC008), the National Natural Science Foundation of China (31801845), and the Natural Science Foundation of Shandong Province (ZR2019BC075, ZR2020MC141, ZR2021MC177). The funders had no role in study design, data collection and analysis, decision to publish, or preparation of the manuscript.

==============================
Exogenous melatonin (MT) is widely used in fruit preservation, and can increase the storage time and delay the quality deterioration. Firstly, it was found that 150 μM MT was the optimal concentration to treat ‘Xinli No.7’ under storage at 4 °C for 60 days. MT could significantly improve oxidase activity and inhibit the reduction of physiological indexes, including pulp hardness, weight loss, titratable acid and soluble solid content. MT could also reduce ethylene release and limit the reduction of fruit aroma. The average content of fruit aroma substance increased by 43.53%. A relevant RNA-Seq database was built to further explore the regulation mechanism of MT. A total of 2,761 differentially expressed genes (DEGs) were identified. DEGs were enriched in 64 functional groups and 191 Kyoto Encyclopedia of Genes and Genomes (KEGG) pathways. DEGs were mainly enriched in alpha-linolenic acid metabolism, fatty acid metabolism and plant hormone signal transduction pathway. The gene pycom09g05270 belonging to long chain acyl-CoA synthetase family and participating in fatty acid metabolism pathway was identified, and its expression level was consistent with fragments per kilobase per million mapped reads (FPKM) values, implying that pycom09g05270 might play a vital role in maintaining quality during the storage process.

Introduction

Pear (Pyrus) is one of the three largest deciduous fruit trees in the world, and fruit aroma is one of the important indexes to evaluate the quality of pear fruit and determine consumers’ preference. Occidental pear varieties have richer fruit aromas as compared to Oriental varieties. The fruit aroma and storage are affected by some irreversible physiological and biochemical reactions occurring inside the fruit along with fruit ripening and development (Jia et al., 2018). On the other hand, the pear fruit aroma would become weak under low temperature storage, which leads to a decrease in the edible quality and commercial value.

Exogenous melatonin (MT) (N-acetyl-5-methoxytryptamine) can limit the decrease in fruit aroma during storage and prolong the storage time. MT is a kind of indole small molecule substance which is widely spread in animals and plants (Jang, Zu & Center, 2015; Wang, Yang & Li, 2016; Jemima, Bhattacharjee & Singhal, 2011), and was discovered in the pineal gland of cattle. Initial research held that the substance only existed in animals and humans. Until 1995, Hattori, Migitaka & Iigo (1995) found that MT was also present in some plants, such as wheat and corn. Afterwards, MT was discovered in roots, leaves and flowers of plants (Hattori, Migitaka & Iigo, 1995).

Studies have shown that MT can scavenge reactive oxide species (ROS) (Lei, Wang & Tang, 2013; Pieri, Moroni & Marra, 1995), regulate fruit maturity and senescence (Gao, Zhang & Chai, 2016; Xin, Si & Kou, 2017; Hu, Li & Rao, 2018) and improve stress resistance (Lei et al., 2010; Jia et al., 2019). In addition, MT has also been proven to have a better preservation effect on fruits and vegetables. In peach, MT treatment could reduce decay rate, delay senescence and maintain total soluble solids and ascorbic acid content during storage (Gao, Zhang & Chai, 2016). A total of 0.2 mM MT could delay ripening, maintain and improve quality of mangoes via inhibiting hydrolytic enzymes and enhancing the antioxidant system (Awad & Al-Qurashi, 2021). In the apple ripening process, the malondialdehyde (MDA) content was negatively correlated with the MT content. This may be due to the fact that MT can remove the ROS produced by the respiratory jump to maintain the redox balance in the cell (Lei, Wang & Tang, 2013). Additionally, MT could regulate ROS signaling followed by activation of the calcineurin B-like 1-interacting protein kinases 23 (CIPK23) pathway to regulate the expression of the potassium channel protein gene, which then promotes K+ absorption (Li et al., 2016). MT could inhibit ethylene biosynthesis to alleviate O3 stress in grape leaves (Liu et al., 2021).

The ability of MT to scavenge radicals is higher than that of vitamin E, ascorbic acid and glutathione, and it is an efficient redox balance agent (Pieri, Moroni & Marra, 1995). In cucumber, MT could inhibit the respiration rate and reduce ethylene release, MDA content and active oxygen content during storage (Xin, Si & Kou, 2017). The carrot suspension cell line treated with MT could better maintain the function and integrity of the cell membrane under low temperature stress (Jia et al., 2019). A total of 100 μmol·L−1 exogenous MT treatment could alleviate the decrease in chlorophyll, ascorbic acid (Vc), titratable acid and soluble protein contents, inhibit membrane lipid peroxidation and maintain membrane function and integrity of cucumber after harvest (Xin, Si & Kou, 2017). In cabbage tissues stored at low temperature after harvest, MT could increase the activity of superoxide dismutase (SOD) and peroxidase (POD), reduce MDA accumulation, delay chlorophyll degradation and maintain soluble sugar and soluble protein contents in stem and leaf tissues (Jia et al., 2019). In plum fruits, suitable concentration of MT treatment could keep the contents of fruit soluble solids and titratable acid stable and ensure the fruit flavor and quality (Feng et al., 2020). MT could inhibit phenolic compound metabolism and improve antioxidant capacity to delay the surface discoloration of fresh-cut Chinese water chestnuts and prolong the shelf life (Xu et al., 2022).

MT can induce ethylene synthesis and signal transduction, inhibit the accumulation of free radicals and membrane lipid peroxidation and promote fruit maturity and delay fruit senescence. However, research on the effect of MT on pear fruits are few and mainly focus on the quality. MT can participate in antioxidant and antibacterial process in ‘Huangguan’ pear during storage. Liu (2019) studied the effects of 100 µmol·L−1 MT on the softening and senescence of three Western pear cultivars after low temperature storage and further analyzed aroma components change of ‘Korla pear’ and ‘Abate’. However, the suitable concentration of MT treatment in the process of low temperature storage was not explicably stated.

Until now, the molecular mechanism of MT in regulating pear fruit aroma is still unclear. In this study, six MT concentrations were used to treat ‘Xinli No.7’ fruits before storage at low temperature. The pulp hardness, weight loss rate, soluble solid content, titratable acid, MDA, POD, SOD, ethylene release rate, fruit aroma content and aroma components were measured at different storage days. Treating the fruits of ‘Xinli No.7’ with 150 µmol·L−1 MT can significantly maintain the physiology quality of pear and delay aroma decrease during low temperature storage. Gene expression types and abundance of MT-treated pear fruits during low temperature storage were further analyzed using digital expression profiling technology. Hence, the key genes involved in regulating aroma production under low temperature were identified and the molecular mechanism of MT in the regulation of fruit aroma was explored.

Materials and Methods

Experimental materials

‘Xinli No.7’ is one of the filial generations of ‘Korla pear’ × ‘ZaoSu’, which belongs to P. sinkiangensis and is a new early-maturing pear variety. Additionally, the scientific name is P. sinkiangensis Xinli No.7.

Xinli No.7 fruits were collected from Tianping Lake Experimental Demonstration Base of Shandong Institute of Pomology. The pear fruits with uniform maturity, uniform size and without mechanical damage were selected and divided into six groups. Six concentrations (0, 50, 100, 150, 200 and 250 µmol·L−1) of MT solution were set to treat six pear groups. The pears were immersed in MT solution for 30 min, and then dried in natural conditions and stored at 4 °C. Afterwards, the physiological indicators were determined at 20, 40, 60 and 80 days after storage. Additionally, the MT was purchased from Solarbio (Cas:73-31-4) and dissolved with dimethyl sulfoxide (DMSO).

Analysis of physiological indexes

The fruits treated with different MT concentrations and stored at 4 °C were taken out every 20 days to measure physiological data and enzyme activity.

Pulp hardness was measured using a GY-4 hardness tester. The skin was peeled off the carcass of the fruit, and the hardness of the sunny side and the dark side of each fruit was measured.

The weight loss rate of pear under treatment with MT and storage at 4 °C were calculated using the following formula: weight loss rate (%) = (initial weight − post storage weight)/initial weight × 100. Firstly, the weight of MT-treated pear before storage was recorded, and the same pears were taken after storage for 20, 40, 60 and 80 days to weigh and record the weight. Then, the weight loss rate was calculated.

The soluble solid contents of pear were measured using ATAGO-PAL-Q type digital refractometer. The detection mirror was cleaned with distilled water, then adjusted to zero for calibration, and the detection mirror was dried with lens wiping paper. Squeezed juice was dropped on the detection mirror of the refraction instrument for measurement, and the value on the digital display device was read. Each treatment was repeated three times.

The titratable acid content of fruits was determined using the sodium hydroxide titration method. Firstly, 5 g pulp was weighed and grinded into homogenate in a mortar, and transferred into 50 ml conical bottle. The solution was placed in a water bathed for 30 min at 80 °C, and then cooled to room temperature. Afterwards, the solution was centrifuged at 10,000 g for 10 min, and the precipitation was removed. The supernatant was added into a 50 ml volumetric flask and then distilled water was added into it until it reached 50 ml. The supernatant solution was blended, and 10 ml supernatant solution was taken for determination of acid content. Total of ~3–5 drops of phenolphthalein indicator were added into the 10 ml supernatant solution, and 0.10 mol·L−1 NaOH standard liquid was added until the solution turned red and did not fade for 30 s. Each treatment was repeated three times. The titratable acid content of pear under treatment with MT and storage at 4 °C was calculated using the following formula: titratable acid content (%) = (C × V1 × k × VT)/m × VS. C, V1, k, VT, m and Vs represent concentration of NaOH standard solution (mol·L−1), the volume of NaOH standard liquid (ml), organic acid conversion factor, constant volume after extraction (ml), sample quality (g) and volume of sample solution used for titration (ml), respectively.

Analysis of enzyme activity

Firstly, 2 g pulp was weighed and put into a precooled mortar. A total of 5 ml pre-cooled 100 mmol· L−1 phosphoric acid buffer with pH 7.0 was added into the mortar, and then, a little of 1% polyvinylpyrrolidone (PVP) and a small amount of quartz sand were added. The homogenate was ground in an ice bath and centrifuged at 10,000 rpm and 4 °C for 20 min. The supernatant was the crude enzyme extract for determination of malondialdehyde content and antioxidant enzyme activity.

MDA content was measured using the thiobarbituric acid (TBA) method. The solution of 0.6%TBA (2 ml) was added to the supernatant (2 ml) and the water was used as references, respectively. The solution was mixed and water bathed at 100 °C for 15 min, and then, they were quickly cooled and centrifuged at 10,000 rpm and 4 °C for 20 min. The absorbance value of the supernatant was determined at 450, 532 and 600 nm, respectively. MDA content (mmol·g−1) = [6.45(D532 − D600) − 0.56D450] × N/W. N and W represented the volume of extract solution (ml) and the weight of pulp (g), respectively.

SOD activity was measured using the azblue tetrazole photochemical reduction process method. Firstly, reaction reagents, including phosphate buffer, Met solution (13 mmol), NBT solution (75 μmol), EDTA-Na2 (10 μmol), riboflavin (2.00 μmol) and distilled water were prepared. Four mixed solutions (2.9 ml) were prepared, and two fractions of mixed solutions were added to the enzyme extract (0.1 ml) and the others were added to phosphate buffer as controls. One of the two controls was covered with foil. They were placed in 4,000 l× sunlight lamp for 20 min at 25 °C. They were quickly transferred into dark condition to stop the reaction. The absorbance value for reaction solutions was determined at 560 nm. SOD enzyme activity = (A0−As) × Vt/C × 0.5A0 × Fw × V1. A0, As, Vt, V1, Fw and C represented the absorbance value of the control, the absorbance value of the sample, the total volume of enzyme extract solution (ml), the sample volume used in testing (ml), the weight of sample (g) and protein concentration in crude enzyme solution (mg/g), respectively.

POD activity was measured using the guaiacol method. The reaction buffer was prepared, including 0.05 mol/L guaiacol solution, 2% H2O2, 20% TCA, and 0.1% PBS. The enzyme extract solution and PBS (control) were added to the reaction buffer (3 ml), respectively. The absorbance value of the reaction solutions was determined at 470 nm and immediately the values were recorded every 10 s, three times. POD enzyme activity (U/g min) = ΔA470 × Vt/0.01W × Vs × t. ΔA470, W, t, Vt, and Vs represented change of absorbance value in reaction time, the weight of sample (g), reaction time (min), the total volume of enzyme extract solution (ml) and the sample volume used in testing (ml), respectively.

Analysis of ethylene release rate

The pear fruits were placed in a dry container for 2 h after having been sealed. A 1 ml syringe was used to draw air repeatedly and the air sample was injected into the GC2014C gas chromatograph for determination. The FID detector was used and the test parameters were set as follows. The N2 flow rate was 85 ml/min, H2 pressure was 0.60 kg/cm2, air pressure was 0.40 kg/cm2, column and injection port temperature were 105 °C and 125 °C, respectively.

Determination of the volatile aroma

The pear fruit were removed from the kernel and cut into 0.20 × 0.20 × 0.20 cm chunks. After sample mixing, 5 g samples were taken and put into a 10 ml sample bottle. A total of 3 µl ance-octanol 2 (0.01644 g·L−1) was dropped into the sample bottle and covered with tin foil seal. The methods of fruit extraction and determination of volatile components were as follows. The sample was water bathed for 30 min at 40 °C. Meanwhile, the aroma were adsorbed using a SPME extraction head (black). And then, adsorption components were injected into a Gas Chromatography/Mass Spectrometry (GC-MS) for 5 min. The unknown compounds were searched by Internet and matched with mass spectrometry libraries NIST17-1, NIST17-2 and NIST17-3 to confirm all volatile components. The content of each component was calculated according to the following formulae. Each component (ng·g−1) = [the peak area of component/(the peak area of internal standard × m)] × C × V × 1,000. Among them, m, C and V represented the quality of sample (g), internal standard concentration (g/L) and internal standard volume (μL), respectively.

The transcriptome materials and RNA extraction

The ‘Xinli No.7’ pear fruits were divided into six groups, and each group contained five pears. Three of these groups were treated with 150 μmol·L−1 MT (MT150), and the others were treated with water (CK). The pear fruits were stored at 4 °C for 60 days. The pulp of six groups of pear fruits were collected and placed in a freezer tube. They were put into liquid nitrogen and stored at −80 °C.

The final cDNA libraries were synthesized by Shanghai OE Biotech Co., Ltd. Total RNA was extracted using the mirVana miRNA Isolation Kit (Ambion, Austin, TX, USA) referring the manufacturer’s protocol. Agilent 2,100 Bioanalyzer was used to tested RNA integrity, and the libraries were constructed using TruSeq Stranded mRNA LTSample Prep Kit following the manufacturer’s instructions. Illumina sequencing platform (Illumina HiSeq × Ten) was used to sequenced these libraries, and 125 bp/150 bp paired-end reads were generated.

Transcriptome data analysis

The published genome database of Pyrus communis (taxid: 23211) was used as a reference to analyze the sequencing results. Raw data (raw reads) were processed using Trimmomatic. The reads containing ploy-N and the low-quality reads were removed to obtain the clean reads. Then the clean reads were mapped to Pyrus communis genome using hisat2. FPKM value of each gene was calculated using cufflinks, and the read counts of each gene were obtained by htseq-count. DESeq was applied to carry out the elimination of biological mutations, and the threshold for significantly differential expression were P value < 0.05 and fold change >2 or <0.5. The analysis methods of Hierarchical cluster, GO enrichment and KEGG pathway enrichment analysis were collected as previously described (Bolger, Lohse & Usadel, 2014; Kanehisa et al., 2008; Anders & Huber, 2013). Additionally, the statistical power of this experimental design was calculated in bioconductor (version 3.15), an open-source software for bioinformatics (https://doi.org/doi:10.18129/B9.bioc.RNASeqPower). We also used the software to analyze RNA-seq data, and 10 × sequencing depth and three technical replicates were used to achieve the claimed statistical power. Additionally, the raw sequence data and reads of RNAseq assembles sequences had been deposited in NCBI, and the SRA and TSA accession numbers are PRJNA861374 (TaxID: 870726) and GKBH00000000, respectively. The actual sequences could be downloaded from Genome Database for Rosaceae (GDR (rosaceae.org)).

qRT-PCR analyses

Based on the analysis of RNA-seq data, 10 genes relating to aroma expression were selected to validate their expression levels. The primers were designed by Shanghai OE Biotech Co., Ltd and synthesized by TsingKe Biotechnology Technology. The primers were shown in Table S1. Actin was used as an internal control gene. qRT-PCR analysis was carried out using Roche 480 SYBR GREEN Master (Roche, Basel, Switzerland). Roche Light-Cycler software was used to analyze the original data, and the gene expression was standardized to the actin gene to minimize the difference of cDNA template levels (Chen et al., 2015). The qRT-PCR data were shown in three technical repeats with error bars, representing the mean value ± standard deviation (n = 3). The relative expression level was calculated using the 2−ΔΔCt method (Livak & Schmittgen, 2001).

Statistics analysis

Microsoft Excel 2010 was used to summarize data and SPSS16.0 software was used to analyze statistical significance. Data were expressed as means ± standard deviation (SD). One-way ANOVA were used to analyze the normality followed by Duncan’s multiple range test to obtain the statistical significance between groups. Alphabet indicated that data showed significant difference at P < 0.05 level.

Results

Physiological indexes analysis of pear fruit under treatment with MT

In the process of low temperature storage, the fruit became soft and the content of soluble solid decreased, which was one of the important reasons for the decline in fruit quality. To explore the change of physiological indexes of ‘Xinli No.7’ treated with MT during low temperature storage, six concentrations of MT were set to treat ‘Xinli No.7’ pear fruits, these included: 0, 50, 100, 150, 200 and 250 μmol·L−1. MT treatment could significantly inhibit the decrease of fruit hardness along with the extension of storage time (Fig. 1A). MT could inhibit the increase of weight loss during the process of storage. When compared with each group treated with different MT, 150 µmol·L−1 MT could specifically circumvent fruit weight loss (Fig. 1B). In the CK group, soluble solid contents increased until 40 days, and then, it slightly came down. When compared with CK, MT could increase soluble solid contents during storage, and 150 µmol·L−1 MT was beneficial in maintaining and increasing soluble solids in pear fruit (Fig. 1C). Titratable acid content decreased significantly during storage, and MT treatment could slow down the decline of fruit titratable acid content. Compared with other groups treated with MT, 150 µmol·L−1 MT could significantly maintain titratable acid content (Fig. 1D). Therefore, 150 µmol·L−1 MT could not only prolong the storage time of fruits but also maintain physiological indexes of pear.

Figure 1 Physiological indexes analysis of ‘Xinli No.7’ treated with MT.

Including pulp hardness (A), weight loss rate (B), soluble solid contents (C) and titratable acid contents (D). Note: (a–f) indicated that data showed significant difference at P < 0.05 level.

Biochemical indexes analysis of pear fruit under treatment with MT

Fruit physiological and biochemical indexes are closely related to fruit quality. MT can remove ROS, and regulate fruit ripening and senescence, thus prolonging fruit storage time and maintaining fruit quality. In the CK group, MDA contents increased along with the extension of storage time. MT treatment could significantly delay the rise of MDA content (Fig. 2A). In the CK group, POD activity significantly increased until 40 days storage, and then kept at a similar level. Interestingly, POD activity increased under treatment with MT. In the group treated with 150 µmol·L−1 MT, POD activity increased significantly higher compared with other concentrations of MT. Additionally, there was no difference in POD enzyme activity between the control group and the high MT concentration (200 and 250 µmol·L−1) groups at the later stage of storage (Fig. 2B). SOD activity increased during the storage period. MT could significantly improve SOD activity, especially the group treated with 150 µmol·L−1 MT (Fig. 2C).

Figure 2 Biochemical indexes analysis of ‘Xinli No.7’ treated with MT.

Including MDA contents (A), POD activity (B), SOD activity (C) and Ethylene release rate (D). Note: (a–e) indicated that data showed significant difference at P < 0.05 level.

Ethylene participates in the fruit ripening process, and its content is directly related to the formation and release of fruit aroma. The results showed that the ethylene release rate of ‘Xinli No.7’ without MT treatment continued to increase during low temperature storage. The ethylene release rate of ‘Xinli No.7’ treated with different MT concentrations was significantly reduced along with the extension of storage time. After 80 days of low temperature storage, the ethylene release rate of pear fruits treated with 50, 100, 150, 200 and 250 μmol·L−1 MT decreased by 19.02%, 33.48%, 59.40%, 41.62% and 21.90%, respectively. Additionally, after 40 days of storage, the ethylene release rate gradually decreased along with the increase of MT concentration. After 60 and 80 days of storage, the ethylene release rate of pear treated with 150 μmol·L−1 MT could be significantly inhibited than that treated with 200 and 250 μmol·L−1 MT (Fig. 2D).

Aroma substances variation of ‘Xinli No.7’ fruits during storage at 4 °C

In order to explore the changes of aroma substances under MT treatment, ‘Xinli No.7’ fruits treated with different exogenous MT and stored for 60 days at 4 °C were selected to detect their variation of aroma substances. The results showed that pear fruits treated with different MT concentrations had a significant increase in total amount of aroma substances. However, there was a slight difference in the types of aroma substances. The treatments with five MT concentrations could increase the types of aroma substances of ‘Xinli No.7’ except 250 μmol·L−1 MT. Thirty-three aroma substances were detected in the control group, and 36, 33, 35, 34 and 26 were detected in fruits treated with 50, 100, 150, 200 and 250 μmol·L−1 MT, respectively. Additionally, we also tested the aroma of ‘Xinli No.7’ treated with MT and storage for 80 days (Table S2). The results showed that the aroma contents and quantity decreased compared with CK, except that treated with 200 μmol·L−1 MT. This may be due to the fact that after storage for 60 days, the ethylene release rate of pear treated with 200 μmol·L−1 was higher than 150 μmol·L−1, implying the storage quality of the pear fruit might begin to decline. Hence, we focused on studying the aroma contents of pear treated with different MT and storage for 60 days.

The detected aroma substances were mainly esters, alcohols, aldehydes, ketones, and other types of substances. The application of exogenous MT with different concentrations could change types of aroma substances, and the kinds of esters, alcohols and aldehydes reduced along with storage. Exogenous MT treatment can promote the formation of aroma substances, including hexyl acetate, 1-Hexanol, Hexanal, (E)-2-Hexenal, O-xylene and Longicyclen. Meanwhile, it could inhibit the formation of aroma substances such as 1,2,4-benzenetricarboxylic acid, cyclic 1,2-anhydride, nonyl ester, 1-Pentanol, 1-Heptanol, Nonanal, Decanal, 2-Octanone, Ethylbenzene, 13-Methyl tetradecanal and Methyl heptenone (Table 1). This also further implicated the complexity of the MT regulatory mechanism.

Table 1 Aroma substance of ‘Xinli No.7’ treated with MT and stored at 4 °C for 60 days.

Volatiles name	Aroma content (ng·g−1)	
CK	50 μmol·L−1	100 μmol·L−1	150 μmol·L−1	200 μmol·L−1	250 μmol·L−1	
Esters							
Ethyl acetate	1.005 ± 0.155	1.197 ± 0.106	4.213 ± 0.318	0.592 ± 0.042	ND	ND	
Butyl acetate	24.607 ± 1.734	24.612 ± 1.827	22.112 ± 1.283	11.128 ± 0.738	14.828 ± 0.772	20.335 ± 1.428	
Acetic acid, pentyl ester	0.051 ± 0.007	1.116 ± 0.102	1.236 ± 0.123	0.892 ± 0.092	1.243 ± 0.113	ND	
Hexyl acetate	ND	5.349 ± 0.427	6.198 ± 0.416	4.321 ± 0.066	4.962 ± 0.284	4.255 ± 0.248	
n-Propyl acetate	0.264 ± 0.014	0.916 ± 0.042	0.725 ± 0.041	0.709 ± 0.06	0.607 ± 0.046	ND	
Isobutyl acetate	0.091 ± 0.011	0.395 ± 0.016	0.303 ± 0.022	0.27 ± 0.011	0.19 ± 0.062	ND	
Isoamyl acetate	0.365 ± 0.02	1.451 ± 0.172	5.732 ± 0.424	ND	1.373 ± 0.104	0.971 ± 0.094	
Allyl decyl carbonate	ND	0.153 ± 0.022	ND	ND	ND	ND	
1,2,4-Benzenetricarboxylic acid, cyclic 1,2-anhydride, nonyl ester	0.035 ± 0.005	0.014 ± 0.002	ND	ND	ND	ND	
2,2,4-Trimethyl-1,4-pentanediol diisobutyrate	ND	0.093 ± 0.011	0.258 ± 0.016	0.145 ± 0.014	ND	ND	
Hexanedioic acid, 3-methyl-, dimethyl ester	0.003 ± 0.001	ND	ND	ND	0.003 ± 0.001	ND	
(Z)-3,7-Dimethyl-2,7-octadien-1-ol, propanoate(ester)	ND	ND	ND	1.672 ± 0.072	ND	ND	
(all-Z)-5,8,11,14-Eicosatetraenoic acid, methyl ester	ND	ND	ND	ND	0.286 ± 0.02	ND	
Alcohols							
Ethanol	26.283 ± 1.937	ND	ND	58.502 ± 4.827	ND	ND	
Dipentaerythritol	ND	ND	ND	0.005	0.042 ± 0.006	ND	
(Z)-2-hexenol	ND	ND	ND	ND	ND	0.246 ± 0.023	
1-Butanol	ND	ND	0.595 ± 0.024	0.517 ± 0.081	ND	ND	
1-Butanol, 2-methyl-, acetate	0.716 ± 0.036	3.773 ± 0.124	ND	ND	3.697 ± 0.283	1.423 ± 0.133	
1-Pentanol	0.182 ± 0.009	ND	ND	ND	ND	ND	
1-Hexanol	2.208 ± 0.204	10.133 ± 0.631	6.417 ± 0.381	12.442 ± 0.882	8.431 ± 0.502	6.852 ± 0.253	
1-Heptanol	0.453 ± 0.026	0.191 ± 0.033	ND	ND	ND	ND	
2-Methyl-1-butanol	0.044 ± 0.006	0.063 ± 0.023	ND	ND	ND	ND	
2-Ethyl-1hexanol	0.221 ± 0.013	ND	3.251 ± 0.081	ND	ND	1.566 ± 0.163	
2-Octanol	9.864 ± 0.849	9.864 ± 0.473	9.864 ± 0.371	9.864 ± 0.183	9.864 ± 0.542	9.864 ± 0.327	
2,5-Monomethylene-l-rhamnitol	ND	0.092 ± 0.029	0.03 ± 0.003	ND	0.039 ± 0.004	0.033 ± 0.006	
2-Buten-1-ol, 3-methyl-, acetate	ND	0.384 ± 0.041	ND	ND	0.405 ± 0.033	ND	
1-Octyn-3-ol, 4-ethyl-	ND	2.339 ± 0.213	ND	ND	2.334 ± 0.224	ND	
6-Methyl-2-heptanol	0.029 ± 0.004	ND	ND	ND	ND	0.789 ± 0.052	
Aldehydes							
Acetaldehyde	0.85 ± 0.034	ND	ND	2.952 ± 0.182	ND	ND	
Hexanal	27.102 ± 2.011	63.446 ± 4.826	89.382 ± 5.927	55.053 ± 5.001	75.662 ± 6.264	71.032 ± 5.927	
Heptanal	0.257 ± 0.022	0.835 ± 0.036	0.94 ± 0.052	0.662 ± 0.089	0.947 ± 0.093	0.273 ± 0.031	
Nonanal	0.892 ± 0.042	0.259 ± 0.026	0.463 ± 0.032	0.22 ± 0.021	0.338 ± 0.022	0.122 ± 0.014	
Decanal	0.355 ± 0.027	0.11 ± 0.017	0.061 ± 0.004	ND	ND	ND	
(E)-2-Hexenal	2.942 ± 0.304	18.168 ± 1.273	15.651 ± 0.992	10.162 ± 0.826	13.415 ± 0.273	5.579 ± 0.123	
2,4,6-Trihydroxybenzaldehyde	ND	ND	ND	ND	0.111 ± 0.012	ND	
3,5-Dimethyl-4-hydroxybenzaldehyde	ND	ND	ND	ND	0.213 ± 0.035	ND	
3-Methyl butanal	0.011 ± 0.002	0.016 ± 0.002	ND	0.016 ± 0.001	0.021 ± 0.003	ND	
13-Methyl tetradecanal	0.102 ± 0.012	ND	ND	ND	ND	ND	
Ketones							
Methyl heptenone	0.167 ± 0.021	ND	ND	ND	ND	0.091 ± 0.015	
Cyclopentanone	ND	0.048 ± 0.007	ND	0.008 ± 0.001	0.011 ± 0.001	ND	
2-Octanone	6.094 ± 0.635	0.441 ± 0.023	0.513 ± 0.052	0.405 ± 0.019	0.349 ± 0.03	ND	
Others							
Toluene	0.375 ± 0.028	0.816 ± 0.041	0.525 ± 0.031	0.782 ± 0.057	0.397 ± 0.036	ND	
Ethylbenzene	1.27 ± 0.052	ND	ND	ND	ND	ND	
O-xylene	0.974 ± 0.034	2.152 ± 0.22	2.517 ± 0.073	1.685 ± 0.163	2.065 ± 0.163	1.09 ± 0.082	
Biphenylene	ND	0.253 ± 0.021	0.515 ± 0.038	0.302 ± 0.068	ND	0.078 ± 0.015	
1,3-Dimethyl benzene	ND	ND	ND	ND	ND	0.854 ± 0.061	
2-Allyl-4-methylphenol	ND	ND	ND	0.391 ± 0.072	ND	0.158 ± 0.016	
Butylated hydroxytoluene	ND	ND	0.505 ± 0.033	0.099 ± 0.011	ND	ND	
1,2-Benzenediol, 3,5-bis(1,1-dimethylethyl)-	ND	ND	ND	0.138 ± 0.042	ND	0.016 ± 0.002	
n-Hexane	ND	0.122 ± 0.017	0.29 ± 0.013	0.15 ± 0.045	0.231 ± 0.032	ND	
Dodecane	0.025 ± 0.003	0.047 ± 0.008	ND	0.032 ± 0.006	0.049 ± 0.007	ND	
Tetradecane	ND	ND	0.27 ± 0.011	ND	ND	ND	
Dodecane, 2,6,11-trimethyl-	ND	ND	0.24 ± 0.018	ND	ND	0.262 ± 0.02	
Nonane, 5-(1-methylpropyl)-	ND	ND	0.017 ± 0.002	ND	ND	0.03 ± 0.001	
Longicyclen	ND	0.025 ± 0.004	0.134 ± 0.011	0.02 ± 0.003	0.026 ± 0.003	0.105 ± 0.022	
Estragole	0.232 ± 0.024	2.011 ± 0.371	0.95 ± 0.056	ND	ND	0.398 ± 0.028	
Chamigren	ND	ND	2.045 ± 0.052	ND	ND	ND	
Valencene	ND	ND	ND	ND	0.207 ± 0.073	ND	
Limonene	ND	ND	ND	ND	ND	1.241 ± 0.094	
acenaphthene	ND	0.184 ± 0.013	0.382 ± 0.021	0.305 ± 0.072	ND	ND	
Fluorene	ND	0.287 ± 0.025	0	0.31 ± 0.067	ND	ND	
Retinoic acid	ND	0.023 ± 0.006	0	ND	0.003	ND	
1,3-Dioxolane, 2,4,5-trimethyl-	ND	ND	0.034 ± 0.008	0.089 ± 0.01	0.059 ± 0.005	ND	
Methyleugenol	0.035 ± 0.005	ND	1.293 ± 0.082	0.818 ± 0.062	0.32 ± 0.036	0.156 ± 0.01	
beta-arone	ND	ND	ND	ND	0.046 ± 0.008	ND	
1,6-dimethylnaphthalene	0.004	ND	ND	0.598 ± 0.061	ND	ND	
Note:

ND means not detected.

Aroma substance contents of ’Xinli No.7’ during storage at low temperature

The content of aroma substance was further analyzed, and it was found that MT could increase the aroma contents of ‘Xinli No.7’ fruit during storage. The total aroma contents of treatment with 0, 50, 100, 150, 200 and 250 μmol·L−1 MT were 108.11, 151.378, 177.661, 176.256, 142.774 and 127.819 ng·g−1, respectively. Compared with control, the total aroma contents treated with MT significantly increased (Fig. 3F).

Figure 3 Comparison of aroma substance contents of ‘XinliNo.7’ treated with different MTs and storage 4 °C for 60 days.

(A–F) Esters, alcohols, aldehydes, ketones, other and aroma total contents of ‘Xinli No.7’ treated with MT. Note: (a–e) indicated that data showed significant difference at P < 0.05 level.

In terms of ester, 50 and 100 μmol·L−1 MT could significantly increase its content, while 100 μmol·L−1 MT could significantly reduce its content, and 200 and 250 μmol·L−1 MT have no significant effect. The content of ester aroma substances of treatment with 100 μmol·L−1 MT was higher (54.33%) than that of control, and 150 μmol·L−1 MT treatment was 25.33% lower than that of CK (Fig. 3A). In terms of alcohols, different concentrations of MT could reduce alcohol contents excepted 150 μmol·L−1 MT. In comparison with control, alcohol contents of samples treated with 150 µmol·L−1 MT increased by 103.33%, and that treated with 100 μmol·L−1 MT decreased by 49.61% (Fig. 3B). Aldehyde content of samples treated with five MT concentrations increased compared with control. Among them, aldehyde contents treated with 100 μmol·L−1 MT increased by 227.57% (Fig. 3C). Interestingly, ketone content treated with exogenous MT were reduced (Fig. 3D). The content of other substances (olefins, acids, etc.) in samples treated with MT significantly increased, and the increase was up to 233.23% when treated with 100 μmol·L−1 MT (Fig. 3E). In summary, exogenous MT treatment can significantly increase the content of pear aroma substances. Both 100 and 150 μmol·L−1 exogenous MT could significantly increase aroma substance content of pear fruits (Fig. 3F), but exogenous MT of 150 μmol·L−1 can significantly increase alcohol content of the fruit.

When compared with the six groups, 150 μmol·L−1 MT was the optimal concentration, and it could significantly reduce the decrease of fruit aroma during the process of long-term low temperature storage. Additionally, it cannot only maintain the storage quality of fruits, but also improve POD and SOD enzyme activity and inhibit MDA contents and ethylene release, thus prolonging the storage time and improving fruits quality.

Differentially expressed genes (DEGs) analysis

In order to further explore the regulation mechanism of MT on ‘Xinli No.7’ fruit aroma under low temperature storage, Illumina sequencing technology was applied to construct the cDNA library. ‘Xinli No.7’ pear fruits were treated with 150 μmol·L−1 MT (MT150) and water (CK) and stored at 4 °C for 60 days. Hence, more than 48 million high-quality reads were obtained in each sample. The reads were compared with pear genome (taxid:23211) (Gareth et al., 2019; Chagné et al., 2014) and the genome coverage rates were higher than 90.3% (Table S3). It showed that total sequencing satisfied the requirement of the downstream analysis. The DEGs were shown by Volcano plot (Fig. 4). In the Volcano plot, P < 0.05 was set as the cut-off criterion of significant difference. Total of 2,761 DEGs were identified from the comparison groups. Additionally, we also analysis the function annotations of 2761 DEGs (Table S4). Most of DEGs were transcription factors (bHLH, WRKY, MADS) and receptor-like serine/threonine-protein kinase (LRR, wall-associated receptor kinase, MLO-like protein), suggesting that DEGs could be involved in multiple process of biology. Among them, 734 genes were up-regulated (orange dots), and 2,027 genes were down-regulated (blue dots).

Figure 4 Volcanic maps analysis of DEGs.

Orange dots and blue dots represented significantly up-regulated genes and down-regulated genes, respectively.

GO annotation analysis of DEGs

GO is an international standard classification system for gene functions. In this study, GO annotations were performed on DEGs of different comparison groups, and DEGs were enriched in 64 functional groups. Among them, 23, 20 and 21 functional groups belonged to three functional family groups, including biological process, cellular component and molecular function. Additionally, the top 10 functional groups of the three functional families were further explored.

A total of 544 DEGs were enriched in biological process. 213, 161, 48, 33 and 32 DEGs were enriched in protein phosphorylation, regulation of transcription, signal transduction, lipid metabolic process and protein ubiquitination functional groups, respectively. A total of 174 DEGs were enriched in cellular component. A total of 117, 14, 11 and 10 DEGs were enriched in membrane, extracellular region, apoplast and cell wall functional groups, respectively. A total of 959 DEGs were enriched in molecular function. Then 390, 213, 119, 73, 59 and 33 DEGs were enriched in ATP binding, protein kinase activity, DNA-binding transcription factor activity, ADP binding, sequence-specific DNA binding and flavin adenine dinucleotide binding functional groups, respectively (Fig. 5; Table S5). Most of the DEGs were distributed in biological process, implying that these genes might play an essential role in the aroma synthesis and pear metabolism under treatment with MT.

Figure 5 GO functional classification of DEGs under treated with MT.

KEGG pathway analysis of DEGs

In organisms, different genes coordinate with each other to perform their biological functions. Pathway enrichment analysis helps to further understand the biological functions of genes. The DEGs were enriched in 191 KEGG pathways based on metabolic pathways (Table S6).

The top 20 KEGG enrichment pathways of total DEGs, up-regulation DEGs and down-regulation DEGs, respectively, were also further explored. A total of 39, 26, 22, 17, 14 and 12 DEGs were involved in plant hormone signal transduction (ko04075), starch and sucrose metabolism (ko00500), endocytosis (ko04144), flavonoid biosynthesis (ko00941), pentose and glucuronate interconversions (ko00040) and alpha-linolenic acid metabolism (ko00592), respectively (Fig. 6). On the other hand, it was found that 13, 12, 12, 12 and 10 up-regulation DEGs were involved in starch and sucrose metabolism (ko00500), flavonoid biosynthesis (ko00941), endocytosis (ko04144), plant hormone signal transduction (ko04075) and protein processing in endoplasmic reticulum (ko04141), respectively (Figure S1). Among them, the KEGG pathways of up-regulation DEGs were nearly consistent with the total DEGs.

Figure 6 KEGG functional classification analysis of DEGs under treated with MT.

A total of 27, 21, 12, 9, 9, 9, 9 and 8 up-regulation DEGs were involved in plant hormone signal transduction (ko04075), biosynthesis of amino acids (ko01230), pentose and glucuronate interconversions (ko00040), sesquiterpenoid and triterpenoid biosynthesis (ko00909), steroid biosynthesis (ko00100), alpha-Linolenic acid metabolism (ko00592), glycine, serine and threonine metabolism (ko00260), fructose and mannose metabolism (ko00051), and phenylalanine, tyrosine and tryptophan biosynthesis (ko00400), respectively (Figure S2). Interestingly, the enrichment KEGG pathways of down-regulation DEGs were significantly different from those of up-regulation DEGs and total DEGs. Additionally, down-regulation DEGs were involved in unique pathways in the top 20 KEGG pathways, including SNARE interactions in vesicular transport (ko04130) and biosynthesis of amino acids (ko01230) (Figures S1 and S2).

The gene expression levels involved in amino acid metabolism and biosynthesis pathways were suppressed, implying that the metabolic processes of pear during low temperature storage might be reduced. On the other hand, the genes involved in fatty acid metabolism, containing alpha-linolenic acid metabolism, linoleic acid metabolism and fatty acid elongation, were up-regulated, implying that fatty acid metabolism was the main pathway of aroma production during low temperature storage. Interestingly, pycom11g01870 (ko00592) and pycom09g05270 (ko02024) belonging to the long chain Acyl-CoA synthetase family were identified, and they were reported to participate in fatty acids β-oxidation process. Although, quorum sensing (ko02024) pathway was not divided into fatty acid metabolism, it might be localized in peroxisome, and participate in transporting fatty acids into peroxisome for the β-oxidation process. Additionally, the FPKM values of pycom09g05270 were significantly higher than those of pycom11g01870. Therefore, it was assumed in the present study that pycom09g05270 might be the key gene in limiting the aroma reduction under MT treatment.

DEGs expression level analysis

To further explore the regulatory mechanism of DEGs, 10 candidate genes related to multiple metabolic pathways from the database based on GO and KEGG analysis were selected to verify their expression trends by qRT-PCR. The 10 genes were involved in fatty acid synthesis pathway, amino acid pathway, transcription factor regulation and calcium signal pathway.

The results showed that the 10 candidate genes shared a similar trend with the FPKM values (Fig. 7). Interestingly, the expression levels of both pycom09g05270 and pycom06g10250 significantly increased under treatment of MT and storage at 4 °C for 60 days, and the rest of the genes showed a downward trend. Based on the above results, it was assumed in the present study that pycom09g05270 might be involved in transporting aroma substrate, and pycom06g10250 might participate in the process of fruit aroma synthesis as a signal.

Figure 7 qRT-PCR analysis of ten DEGs expression level combined with RNA-Seq.

CK and MT indicate that ‘Xinli No.7’ were treated with water and MT, and stored at 4 °C for 60 days.

Discussion

MT treatment could contribute to prolonging the storage time and improving pear aroma, which may be due to a decreased release of ethylene and the delay of respiration jump of pear fruits. In the respiratory jump fruit, aroma synthesis was related to ethylene release. The synthesis of fruit aroma was positively correlated with the release rate of ethylene, and the fruit aroma also began to synthesize in a large amount during the peak period of ethylene release. In pear, the expression peak of PuACS1 and PuACO4 were consistent with the peak of ester aroma release (Li et al., 2014). In mango, the biosynthesis of monoterpenes, esters and aldehydes were strongly dependent on the production of ethylene (Lalel, Singh & Tan, 2003). In melon, the aroma release has also been confirmed to be strongly associated with ethylene (Lucchetta, Manríquez & El-Sharkawy, 2007). Ester biosynthetic pathway was dependent on ethylene because ethylene suppressed fruit had reduced fatty acids and aldehydes (Flores et al., 2002). These reports indicated that ethylene has a significant role in regulating aroma biosynthesis. In jujube, 25 µmol·L−1 exogenous MT could inhibit the fruit respiration and ethylene release, and delay fruit softening (Tang, 2019). However, 50 µmol·L−1 exogenous MT could promote ethylene production and reach the peak of respiratory jump earlier than the control and improve its quality in tomato (Gross, 1985; Sun, 2016). In this study, 150 μmol·L−1 exogenous MT was selected as the optimal concentration to treat ‘Xinli No.7’ fruits, which greatly reduced ethylene release. On the other hand, combined with the results of previous studies, the reduction of ethylene release may also lead to a decrease in fruit aroma, but compared with the control group, the components of fruit aroma increased. This may be due to the fact that MT treatment can significantly increase alcohol contents and reduce ketone contents, increase the concentration of substrates in the process of fatty acid metabolism, thereby limiting the decrease in fruit aroma.

MT could reduce the occurrence of post-harvest fruit diseases, such as pear fruit ring blight, banana anthracnose, jujube penicillium and tomato gray mold (Liu, 2019; Tang, 2019; Li et al., 2019a; Li et al., 2019b). MT enhances fruit disease resistance by enhancing mitogen-activated protein kinase (MAPK) signaling, inducing ROS accumulation and improving the activities of defense enzymes, including chitinase, β-1,3-glucanase, POD, SOD, ascorbic peroxidase (APX) and catalase (CAT). However, MT treatment effectively eliminated defense-related ROS in citrus fruit tissues and accelerated the spread of postharvest green mildew (Wang et al., 2019). On the other hand, MT treatment can improve the resistance of apple (Malus domestica Mill.) leaves to brown spot (Yin et al., 2013). After the tomato bacteria Pseudomonas syringae (Pst DC3000) infection in Arabidopsis thaliana, it could lead to a significant decrease in endogenous MT content. We also screened relevant genes from transcriptome data (Shi et al., 2014). We found that three and 26 DEGs were enriched in regulation of defense response to fungus and defense response function groups, respectively. Among them, the FPKM values of five DEGs were up-regulated, and the others were down-regulated. The expression levels of pycom16g02200, pycom16g13650 and pycom16g00860 were significantly up-regulated under treatment with MT. Interestingly, the expression level of pycom01g16720 was significantly inhibited, implying that the gene might play a key role in defense response (Fig. 5). Additionally, eight DEGs were enriched in MAPK signaling pathway (ko04010), and seven DEGs expression levels were up-regulated (Fig. 6). Therefore, MT may play an important role in plant resistance to fungal diseases.

Through the analysis of transcriptome data, it was found that the expression levels of genes involved in the regulation of fatty acid metabolism have undergone significant changes. The expression level of long chain Acyl-CoA synthetase 8 (pycom09g05270) was up-regulated. The expression levels of linoleate 13S-lipoxygenase 3-1 (pycom09g09820 and pycom09g09830) and allene oxide synthase (pycom08g03050) were down-regulated. It was suggested that fatty acid metabolism may play an essential role in pear fruit aroma metabolism in response to MT treatment during storage. In addition, LACS played an important role in the process of fatty acid transport and activation. Nine LACS genes were identified in Arabidopsis (Shockey, Fulda & Browse, 2002), and fatty acids degradation mainly occurred in peroxidase through the β-oxidation cycle in plants. LACS6 and LACS7 were localized in peroxisomes and expressed in the germination and early seedlings, indicating that they may be involved in the oxidation process of phthalyl-CoA (Fulda, Shockey & Werber, 2002; Fulda, Schnurr & Abbadi, 2004; Zhao et al., 2019). LACS9 was localized in the plasma membrane, and it was inferred that it may be involved in the synthesis of lipids. Additionally, the activity of long-chain acyl-CoA in lacs9-1 mutant was decreased by 90% than wide type (Schnurr et al., 2002). LACS4 and LACS2 also provided substrates for the extension of wax fatty acid precursors (Schnurr, Shockey & Browse, 2004). LACS1 and LACS4 participated in the formation of Arabidopsis pollen coat (Dirk et al., 2011). In the lacs1lacs9 and lacs4lacs9, the fatty acid content decreased by 11% and 27%, respectively (Zhao et al., 2010; Jessen et al., 2015). MdLACS2 could also regulate the accumulation of cuticular wax to enhance plant resistance to abiotic stress and MdLACS4 induced early flowering and increased the adaptability to stress (Zhang et al., 2020a, 2020b). However, LACS involved in β-oxidation process of fruit fatty acids still remained unclear. It was assumed in the present study that during low temperature storage, MT treatment could improve expression level of LACS, which increased the β-oxidation process of fatty acids and the synthesis of substrate, and therefore increased fruit aroma. The mechanism of how fatty acids enter the peroxisome and are activated after entering the peroxisome is still unclear. In conclusion, we could transfer pycom09g05270 into yeast mutants (Δpxa1pxa2faa) to test its critical role in fatty acid transport and analysis β-oxidation capacity of yeast mutant.

Calcium is one of the essential nutrient elements in plants. It is closely related to plant growth and development, and can be used as a signal molecule to regulate the development process. Studies have shown that calcium is one of the inhibitors of ethylene synthesis. Exogenous calcium could reduce ethylene production and delay ethylene peak of respiratory jump fruits. In ‘Nanguoli’, calcium treatment could improve its aroma by increasing the contents of metabolic substrates (linoleic acid and linolenic acid) and the activities of alcohol dehydrogenase (ADH), pyruvate decarboxylase (PDC) and lipoxygenase (LOX) (Wei et al., 2017). Combined with RNA-Seq and qRT-PCR results, pycom06g10250 encoding calcium ions was identified and the expression level was up-regulated under MT treatment. Virus induced gene silencing (VIGS) could be used to further verify the function of pycom06g10250 in promoting pear aroma. This result indirectly showed that in the process of low temperature storage, exogenous MT treatment can activate the calcium ions pathway which could limit the decrease of aroma.

Conclusion

In this article, ‘Xinli No.7’ pear fruits were treated with six concentrations of MT, and MT could contribute to the improvement of oxidase activity, and the inhibition of the reduction of physiological indexes, such as MDA, SOD, POD, etc. 150 μmol·L−1 was also identified as the optimal concentration for treating pear fruits, and it could significantly reduce ethylene release and limit the reduction of fruit aroma during storage. RNA-Seq library was constructed to explore the potential molecular mechanism of MT in slowing down the decrease of fruit aroma. A total of 2,761 DEGs were identified. Among them, 734 DEGs were up-regulated and 2,027 DEGs showed tendency to be downregulated. DEGs were enriched in 64 functional groups and 191 KEGG pathways. The DEGs were involved in plant hormone signal transduction, alpha-linolenic acid metabolism and linoleic acid metabolism pathway. qRT-PCR method was also used to verify the expression trends of DEGs, and it was found that the gene expression patterns were consistent with the FPKM values. The two DEGs, pycom09g05270 and pycom06g10250, belonging to long chain acyl-CoA synthetase 8 and calcineurin B-like protein 7 family, were identified. Their expression levels were up-regulated when treated with MT at low temperature storage, which indirectly indicated their role in the storage process.

Supplemental Information

Supplemental Information 1 KEGG functional classification analysis of up-regulated DEGs under treated with MT.

Click here for additional data file.

Supplemental Information 2 KEGG functional classification analysis of down-regulated DEGs under treated with MT.

Click here for additional data file.

Supplemental Information 3 The primer sequences used in this study.

Click here for additional data file.

Supplemental Information 4 Aroma substance of ‘Xinli No.7’ treated with different concentration of MT and stored at 4 °C for 80 days.

Click here for additional data file.

Supplemental Information 5 List of comparisons between reads and reference genome.

Click here for additional data file.

Supplemental Information 6 The annnoation information of 2761 DEGs.

Click here for additional data file.

Supplemental Information 7 Raw data for Figures 1–3, 7 and Table 1.

Click here for additional data file.

Supplemental Information 8 List of gene ID enrichment in GO functional classification.

Click here for additional data file.

Supplemental Information 9 List of important metabolic pathways of DEGs.

Click here for additional data file.

Additional Information and Declarations

Competing Interests

Author Contributions

Data Availability

The authors declare that they have no competing interests.

Shuwei Wei conceived and designed the experiments, prepared figures and/or tables, and approved the final draft.

Huijun Jiao conceived and designed the experiments, performed the experiments, prepared figures and/or tables, and approved the final draft.

Hongwei Wang performed the experiments, prepared figures and/or tables, and approved the final draft.

Kun Ran analyzed the data, authored or reviewed drafts of the article, and approved the final draft.

Ran Dong analyzed the data, authored or reviewed drafts of the article, and approved the final draft.

Xiaochang Dong analyzed the data, authored or reviewed drafts of the article, and approved the final draft.

Wenjing Yan performed the experiments, authored or reviewed drafts of the article, and approved the final draft.

Shaomin Wang conceived and designed the experiments, prepared figures and/or tables, and approved the final draft.

The following information was supplied regarding data availability:

The raw measurements are available in the Supplemental Files.

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
