# Peer review of "The mechanism analysis of exogenous melatonin in limiting pear fruit aroma decrease under low temperature storage"

_PeerJ, doi:10.7717/peerj.14166_

## Round 0.1 · original submission · Major Revisions

Please respond to the reviewers' comments point by point.

Reviewer 1 ·

Basic reporting

I have carefully read the Manuscript 72082v1. The authors reported the positive effects of exogenous melatonin treatment on post-harvest physiology of pear tree. Furthermore, they also reported the mechanisms involved at the biochemical, hormonal and genetic levels. In my opinion, MS deserves attention to be published.

Experimental design

Well designed and performed.

Validity of the findings

No comment

Additional comments

The authors should discuss the possible effects of melatonin on post-harvest fungal diseases, in light of the genes involved in the defense response.

Reviewer 2 ·

Basic reporting

In this work, the authors determined the effects of melatonin fruit preservation and on the aroma production. RNA-Seq analysis was conduct to revealedthe possible molecular mechanism. However, personally speaking, the methods, data analysis, results and organization should improved a lot. Several suggestions should take into account for improving the manuscript.

Experimental design

The methods should improved greatly, lots of key information missing.

Validity of the findings

The conclusion is too general and should be more clear and detail based on further analysis of the data

Additional comments

Several suggestions should take into account for improving the manuscript.
1,In the abstract, the authors state that that 150μM melatonin was the optimal concentration, but how about the duration? Normally, the treatment effects depend on the does, duration and temperature.
2,All abbreviation should stated the full name at the first time appear, please revised in the abstract and text.
3,MT is the abbreviation of melatonin, but the authors did not mentioned in the text, and change MT and melationin all the time. Please keep uniformly.
4,The introduction seems not logically present, and should improved, and more recently publication about MT referred to your work should cited.
5,‘Xinliqihao’ fruits is what fruit? The scientific name of sample should present.
6,The information about MT should provided and how to prepared it.
7,The methods should improved greatly, lots of key information missing...
8,2.6 The transcriptome materials and RNA extraction. How to take the samples and how many sample for the RNA-Seq analysis?
9,If actin is the most suitable reference gene for your experimental condition? Please proved it.
10,Line 173, the reference should be cited for the 2-ΔΔCt method.
11,Figure 3, which storage time did the aroma determined? More information should provide in the figure legend. And if there is significant difference among different treatments?
12,Why only determined the aroma on 60days, how about on 80 days?
13,The RNA-seq analysis should give more deeply analyze. Too general described of the results and no any key information provided. For example, how about the genes expression in the fatty acid metabolism, amino acid synthesis and plant hormone signal transduction pathway? Most importantly , MT relieve the aroma reduction, how about the key gene for aroma metabolism pathways?
14,The English should improved greatly.

Reviewer 3 ·

Basic reporting

The authors need to carefully check and make corrections throughout the manuscript to improve the English writing style for a scientific report.

Experimental design

no comment

Validity of the findings

some conclusions are not well stated or could not be drawn according to the results.

Additional comments

1. Lines 22-23. 280-281. 348-351. when talking about up- and down-regulated genes, make clear what the comparisons are. Up- and down-regulated terms are relative terms and need to be specified.
2. Line 359, 362. double check spelling and references insertion.
3. Line 98-99. please specify the exact treatment method, immersed or spraying?
4. Line 95. stated the Latin name of the pear fruit, not just its variety name “Xinliqihao”. And it also should not be a keyword in my opinion.
5. Line 311. simply introduce the GO annotations of pycom09g05270 and pycom06g10250 in part 3.8. Line 319-321. and the conclusion is arbitrary just based on the qRT-PCR result.
6. about the unit of mol/L, two forms are present randomly with mol·L-1. The decimals are also random, 0.1, or 0.10, and even 0.1000 (line 113).
7. Line 333-334. “The significant role of ET in delaying the fruit aroma”. Explained how to delay? The conclusion is contradictory to the previous reports.
8. Too many nominal phrases appeared, such as “active oxygen free radicals (line 48)”, “pear fruit aroma decline (line 90)”, “weight loss rate increase (line 187)”, “pear fruit aroma substances variation (line 223)”, ……
9. the ms should be edited by a native English speaker first.
10. Fig 4 has green and red dots in it, and it is not convenient for the daltonism.

---

## Round 0.2 · Minor Revisions

Dear authors: The Section Editor has detected the following issues which should be addressed in the revised version. Please take care of these issues.

There are no connections for the GO and KEGG findings to actual sequences. There is no pointer to the pear reference genome as well. The work provides an overall finding, but does not present any concrete data which would be a foundation for the reader to validate. The work also focuses on a collection of important genes, but does not actually present the sequence data. There is also a need to see the raw data, and RNAseq assembles for the sequencing work which can be deposited in a third-party resource. It is important in some location to list sequences, a coordinate to a reference genome, and the distinct GO: and KEGG annotation for each of the key 2761 DEGs described.

The manuscript also has many language issues, some of which are addressed in the accompanying mark-up copy (to be sent by email). The information once presented will provide a decent manuscript, but there is a major loss of connection to the data here. The manuscript requires considerable revision before further review.

The sequence data needs to be deposited in a third-party resource such as NCBI GenBank, or even Figshare. Assembled transcriptomes can be deposited as a transcriptome shotgun assembly (GenBank TSA resource). Please see: https://www.ncbi.nlm.nih.gov/nuccore/

Reviewer 1 ·

Basic reporting

Authors replied satisfactorily to my suggestion.

Experimental design

Carried out well.

Validity of the findings

Relevant in post-harvest science and technology

Additional comments

MS can be accepted

---

## Round 0.3 · Minor Revisions

The Section Editor, has commented and said:

"The rebuttal mention changes, but they appear missing from the manuscript. There is no mention of the raw data deposition and no mention that focused sequences were obtainable from the GDR resource (mentioned in rebutta, but not manuscript). These are things that cannot be assumed.

The English was improved and passable. What does the 23 pathway statement mean at line 61; it should be clear. Vc mentioned in line 69 is not mentioned until line 137 (and still not defined!). There seems to be a reference missing at line 426 . . . at line 442 it should be Pseudomonas syringae.

In the supplemental tables there should be something to support the figures illustrating the GO assignments like a table with the GO: term (GO:12345) and its text annotation along with the reresentatiove sequence (pycom..).

There is still a need for revision."

Please correct your paper following the suggestions mentioned above carefully!!!

---

## Round 0.4 · Minor Revisions

Please carefully address the statistical issues I explained in my separate email.

---

## Round 0.5 · accepted · Accept

The authors have fully responded to the comments raised by the reviewers and editors and the quality of the manuscript has been improved.